# Attention to audiovisual speech shapes neural processing through feedback-feedforward loops between different nodes of the speech network

**Patrik Wikman** [1,2]*, **Viljami Salmela**[1,2], **Eetu Sjöblom**[1], **Miika Leminen**[1,3], **Matti Laine**[4], **Kimmo Alho**[1,2]

1 Department of Psychology and Logopedics, University of Helsinki, Helsinki, Finland, 2 Advanced Magnetic Imaging Centre, Aalto NeuroImaging, Aalto University, Espoo, Finland, 3 AI and Analytics Unit, Helsinki University Hospital, Helsinki, Finland, 4 Department of Psychology, Åbo Akademi University, Turku, Finland

* patrik.wikman@helsinki.fi

## Abstract

Selective attention-related top-down modulation plays a significant role in separating relevant speech from irrelevant background speech when vocal attributes separating concurrent speakers are small and continuously evolving. Electrophysiological studies have shown that such top-down modulation enhances neural tracking of attended speech. Yet, the specific cortical regions involved remain unclear due to the limited spatial resolution of most electrophysiological techniques. To overcome such limitations, we collected both electroencephalography (EEG) (high temporal resolution) and functional magnetic resonance imaging (fMRI) (high spatial resolution), while human participants selectively attended to speakers in audiovisual scenes containing overlapping cocktail party speech. To utilise the advantages of the respective techniques, we analysed neural tracking of speech using the EEG data and performed representational dissimilarity-based EEG-fMRI fusion. We observed that attention enhanced neural tracking and modulated EEG correlates throughout the latencies studied. Further, attention-related enhancement of neural tracking fluctuated in predictable temporal profiles. We discuss how such temporal dynamics could arise from a combination of interactions between attention and prediction as well as plastic properties of the auditory cortex. EEG-fMRI fusion revealed attention-related iterative feedforward-feedback loops between hierarchically organised nodes of the ventral auditory object related processing stream. Our findings support models where attention facilitates dynamic neural changes in the auditory cortex, ultimately aiding discrimination of relevant sounds from irrelevant ones while conserving neural resources.

## Introduction

Humans effortlessly recognise and separate auditory objects in complex sound environments. This ability relies on hierarchical neural processing in the auditory ventral "what" stream,

**Data Availability Statement:** All data (EEG and fMRI), preprocessed to anonymise, and original code have been deposited at Open Science

Framework under Attention and Memory networks (HTTPS://DOI.ORG/10.17605/OSF.IO/AGXTH). The individual quantitative observations underlying the data summarised in Figs 2B–2D, S1, and S2 is also available in S1–S3 Data files uploaded as a supplement.

**Funding:** This work is supported by the Academy of Finland (grant #297848, "Modulations of brain activity patterns during selective attention to speech", 2016-2020, KA, https://akareport.aka.fi/ibi_apps/WFServlet?IBIF_ex=x_hakkuvaus2&CLICKED_ON=&HAKNRO1=297848&UILANG=en&TULOSTE=HTML) and (grant #1348353 "Solving the puzzle of natural auditory object perception - neural mechanisms in humans and animal models", 2022–2025, PW, https://akareport.aka.fi/ibi_apps/WFServlet?IBIF_ex=x_hakkuvaus2&CLICKED_ON=&HAKNRO1=348353&UILANG=en&TULOSTE=HTML), and the Finnish Cultural Foundation (2020–2025, PW). The funders had no role in study design, data collection and analysis, decision to publish, or preparation of the manuscript.

**Competing interests:** The authors have declared that no competing interests exist.

**Abbreviations:** AV, audiovisual; ECoG, electrocorticography; EEG, electroencephalography; ERP, event-related potential; fMRI, functional magnetic resonance imaging; ICA, independent component analysis; MEG, magnetoencephalography; PE, prediction error; RDM, representational dissimilarity matrix; ROI, region-of-interest; RSA, representational similarity analysis; SER, speech envelope reconstruction; SVM, support vector machine; TRF, temporal response function.

where sequential processing stages extract and integrate increasingly complex object attributes [1,2]—starting with processing of simple features (e.g., frequency) in the primary auditory cortex, progressing to complex acoustic structures (e.g., frequency-modulated sweeps) in secondary areas and selectivity for complete auditory objects in the anterior superior temporal cortex [3–6]. The ventral stream terminates in the anterior temporal and inferior frontal cortex where sound category and semantic information is apparently stored [7–9].

In the absence of spatial cues, there are usually only subtle differences in the vocal attributes that separate concurrent speakers from each other [10]. Therefore, top-down modulation facilitated by selective attention plays a significant role in separating relevant speech objects from irrelevant background speech [10,11]. This top-down modulation is classically assumed to enhance the gain [12–15] or the accuracy [16–18] of responses in neuronal populations processing the relevant sounds. More intricate theories suggest that attention also affects predictive mechanisms in sensory cortices [19] or that attentional modulation arises as neural networks adapt to specific tasks in various contexts [20–23].

Recent methodological advances in electrocorticography (ECoG) [15,24–27], magnetoencephalography (MEG) [11,28], and electroencephalography (EEG) [25,29,30] have revealed that attention enhances neuronal tracking of speech sounds. This amplification is concordant with modulation of both early (i.e., within 100 ms; e.g., [31]) and late (after 100 ms; e.g., [32,33]) neural response curves to sound envelope changes, consistent with the view that selective attention shifts neuronal processing in low-level auditory and higher-level speech-sensitive regions towards the features of the attended speaker [24,31,34,35]. These methods, however, lack spatial precision. That is, ECoG studies are limited by the extent of the implanted electrodes, while MEG/EEG source localisation is relatively inaccurate especially in the case of simultaneously firing neuronal populations [36]. In contrast, functional magnetic resonance imaging (fMRI) provides better spatial resolution, revealing that selective attention to cocktail-party speech modulates information processing in not only low-level auditory regions but also in extensive superior temporal, inferior parietal, and inferior frontal brain regions (e.g., [37–42]). Furthermore, multivariate pattern analyses on cocktail party fMRI data have indicated that neuronal populations that show differential responses during selective attention to speech are distributed globally in disparate cortical regions [38,43]. Yet, fMRI has limitations in estimating the timing of these modulations. Therefore, some fMRI studies have employed a combination of language modelling and multivariate analysis of fMRI responses to address the temporal limitations of fMRI when tracking continuous speech [43,44]. However, here we opted for a different approach by utilising EEG-fMRI fusion [45,46]. This technique allows us to overcome the spatial limitations of EEG and the temporal constraints of fMRI, enabling us to estimate the spatiotemporal characteristics of selective attention to audiovisual (AV) speech.

In the present paradigm, participants watched video clips of dialogues between 2 speakers (dialogue stream) with a distracting speech stream played in the background (background stream; Fig 1A). To increase attentional demands, we modulated the auditory quality of the dialogue stream with noise-vocoding [47] and visual quality in the videos by masking [48]. We also modulated the semantic coherence of the dialogue stream (Fig 1B and 1C). We employed a fully factorial design where participants performed 2 different tasks: (1) attend speech task, where the participants attended to the AV dialogue while ignoring the background speech; and (2) ignore speech task, where the participants ignored both the dialogue and the background speech, and instead counted rotations of a white cross presented visually near the mouth of either speaker. This enabled us to study the effect of selective attention (difference between the attend speech task and the ignore speech task) on both the relevant speech stream (dialogue stream) and the irrelevant background (background stream). We expected that attending to the dialogues would increase speech envelope reconstruction (SER) accuracy of

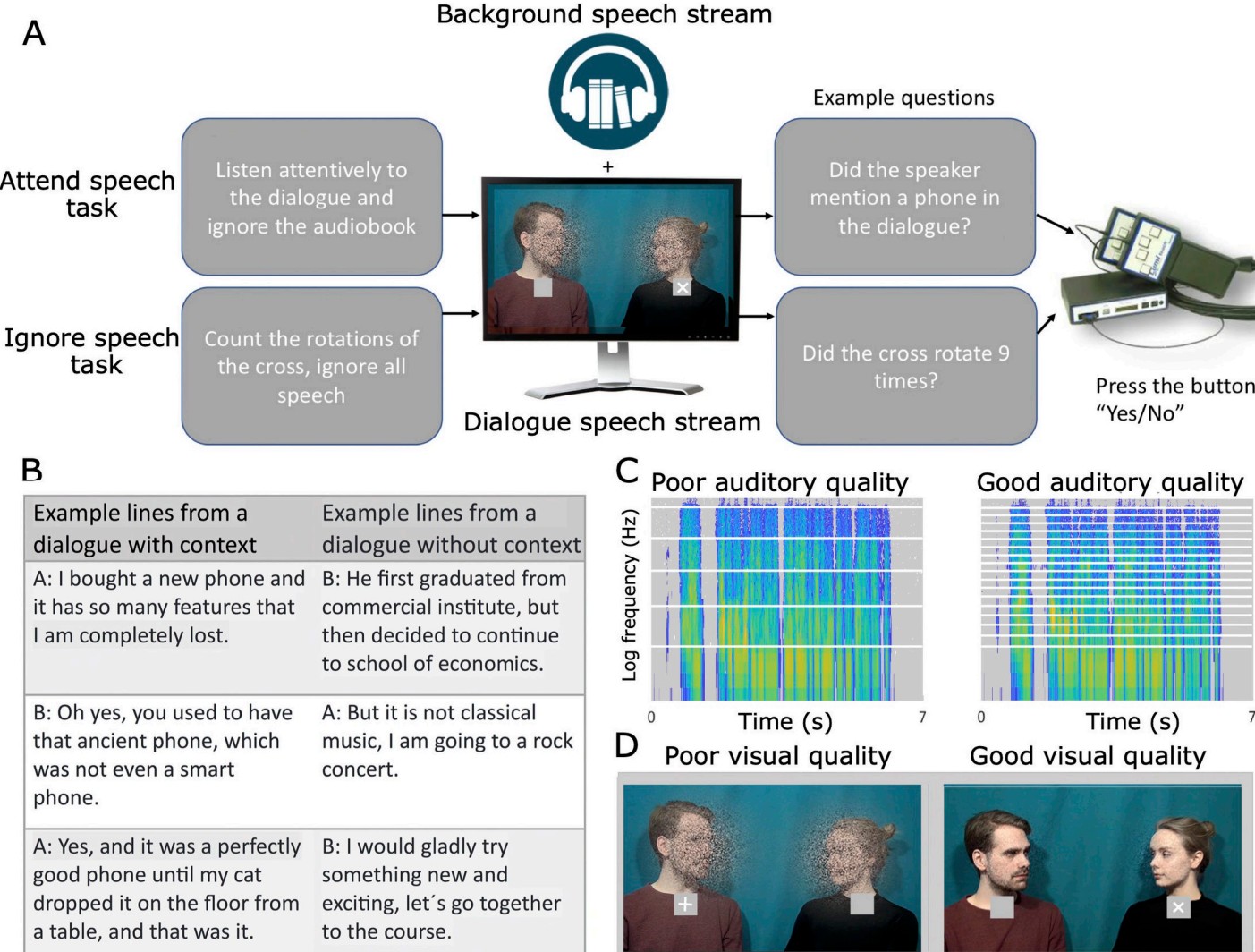

**Fig 1. The AV cocktail party paradigm. (A)** Participants underwent either EEG ($n = 19$) or fMRI ($n = 19$) recordings while watching AV video clips of dialogues consisting of 7 lines (dialogue stream) with a continuous audiobook (background stream) played in the background. Participants performed 2 tasks: (1) an attend speech task where they attended to the dialogue while ignoring background speech; and (2) ignore speech task where they ignored all speech and counted rotations of a cross presented below the neck of the talker. Dialogues were either semantically coherent or incoherent (B), and the audio quality varied with different levels of noise-vocoding (C). Additionally, visual quality was manipulated with dynamic white noise masking (D). AV, audiovisual; EEG, electroencephalography; fMRI, functional magnetic resonance imaging.

the dialogue stream and enhance related early and late neural temporal response components, with opposite effects for the background speech stream. Based on results from our previous fMRI study [38], attentional modulation of the dialogue stream SER accuracy was expected to be temporally variable, changing from line-to-line in a nonlinear fashion. Additionally, we expected SER accuracy to be greatest for dialogues with good audiovisual quality [49,50] and coherent semantics [51]. However, we also wanted to determine whether this applies only to attended speech.

Using SER on the EEG data (Fig 2), we replicated earlier findings that neuronal tracking is amplified for attended speech (see e.g., [29]). Importantly, however, we found that this amplification was not temporally uniform: the tracking amplitude abated linearly with the procession

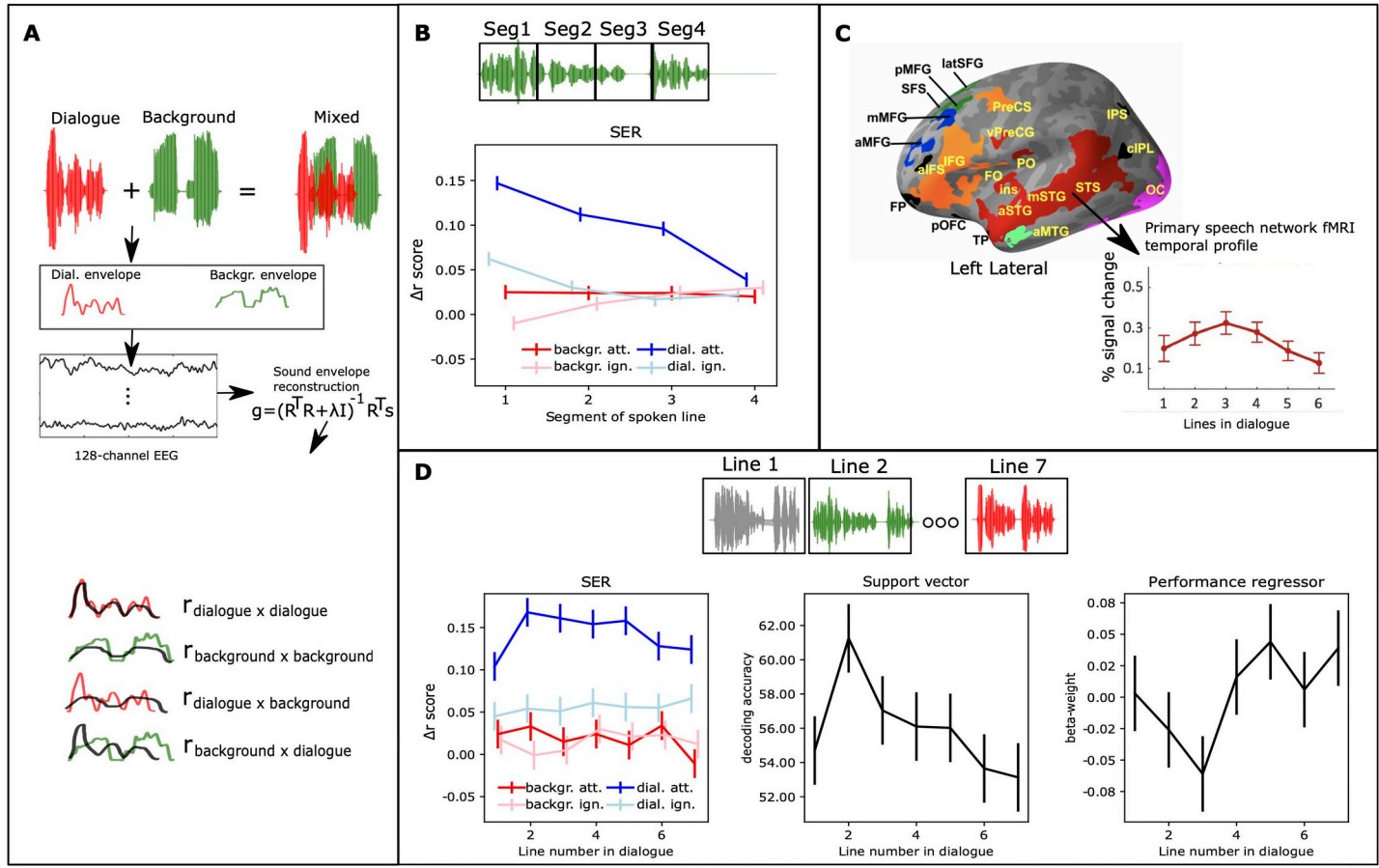

**Fig 2. Schematic illustration of SER and SER results. (A)** Participants heard and saw AV dialogues with overlapping background speech, i.e., a mixed auditory signal. SER was employed to assess neural tracking of the dialogue and background stream. First, we extracted the amplitude envelope for both speech streams. Then, using data from all 128 EEG channels, we separately reconstructed the amplitude envelopes for the dialogue and background stream. To assess the accuracy of neural tracking, we correlated the reconstructed speech with its corresponding envelope and compared this to correlations with the opposite envelope. Accuracy values in B and D represent $\Delta r$ (r-difference scores) between direct correlations and across-reconstruction correlations. **(B)** SER accuracy exhibited a significant linear temporal decrease within each line of the attended dialogue stream. **(C)** Our prior fMRI study [38] demonstrated that attention-related modulation changed from line-to-line in a nonlinear fashion (the red coloured regions, which we named primary speech network in our previous study), the other colours indicate networks where this temporal modulation effect showed another pattern (see [38] for details). **(D)** SER accuracy displayed a similar nonlinear temporal pattern as fMRI (C), but specifically for the attended speech. This trend was observed in both univariate SER accuracy analysis (left) and multivariate SVM decoding (middle; details in "Decoding analysis of SER accuracies"). Participants' SER accuracy was predicted based on their behavioural performance for the attended dialogue stream (right), and this prediction (beta-weight) inversely followed SER accuracy. Error bars indicate ± SEMs. Code and processed EEG data used to generate this figure are archived on the Open Science Framework; HTTPS://DOI.ORG/10.17605/OSF.IO/AGXTH. Data frames are available in S1 Data. AV, audiovisual; EEG, electroencephalography; fMRI, functional magnetic resonance imaging; SER, speech envelope reconstruction; SVM, support vector machine.

of the spoken line. Further, neuronal tracking of attended speech displayed nonlinear fluctuations over the course of the dialogue, similar to those previously reported with fMRI [38]. We discuss how such temporal dynamics may arise due to interactions between prediction and attention and other nonlinear plastic effects in speech processing circuits [19,20]. To evaluate the minute temporal modulation of selective attention, we estimated neural temporal response functions (TRFs) for the EEG data separately for both speech streams (Fig 3). Finally, we performed EEG-fMRI fusion: Based on representational similarity analysis (RSA), we identified brain regions in the fMRI data that contained representational structures similar to those calculated from TRFs, resulting in a TRF-fMRI correlation time series for each brain region (Fig 4 and S1 and S2 Videos, www.mv.helsinki.fi/home/jkaurama/vdialog/, www.mv.helsinki.fi/

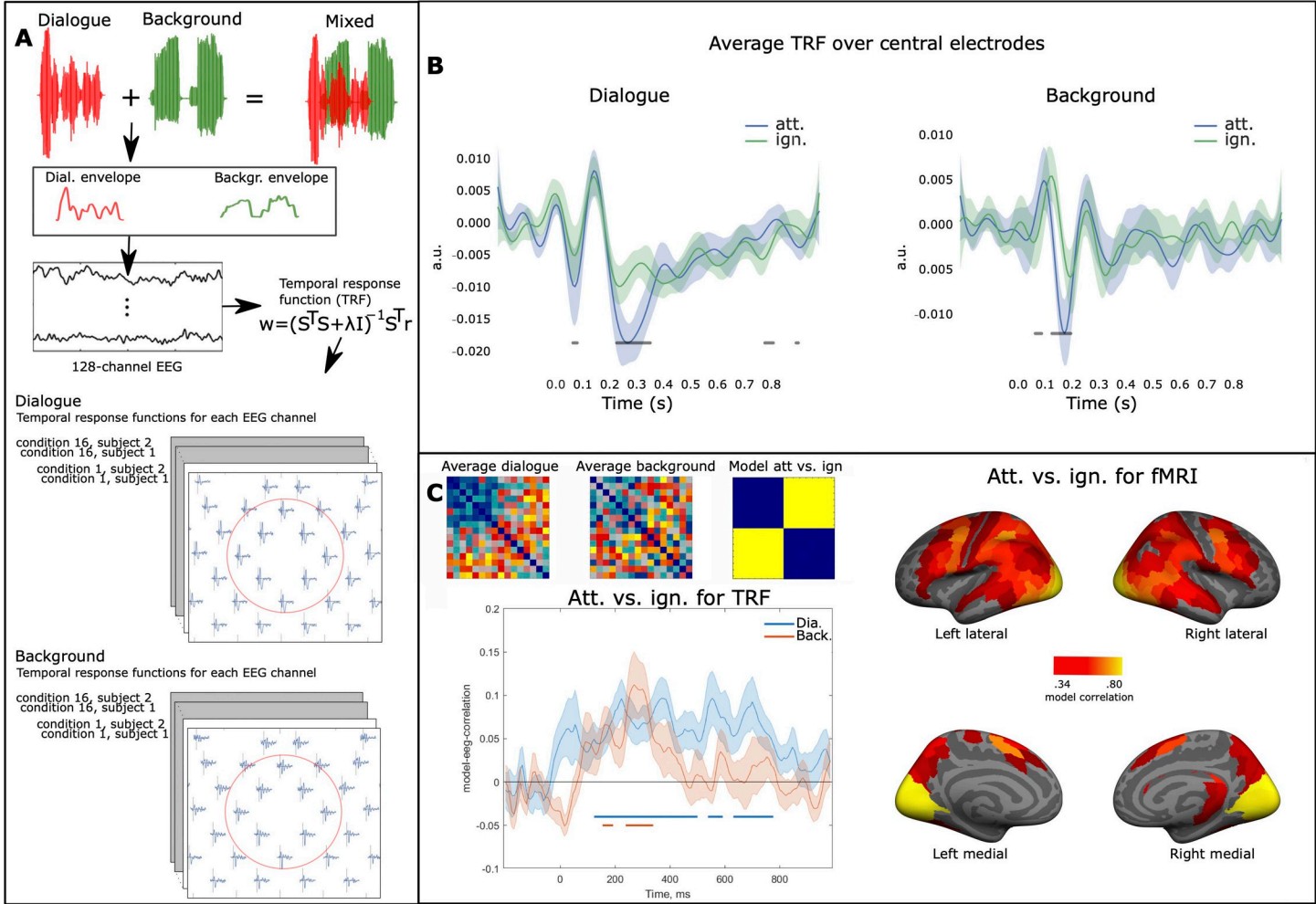

**Fig 3. Schematic of TRF estimation and TRF results.** (A) TRFs were estimated using the same speech amplitude envelopes as in our SER analysis, separately for the dialogue and background streams. (B) Average TRFs over frontocentral electrodes, with points indicating significant differences between the 2 TRFs (paired permutation *t* test *df* = 18, note the 2 streams have separate y-scales). (C) Left: RDMs were constructed using TRFs for all 16 conditions (first the 8 attend speech conditions and thereafter the 8 ignore speech conditions). This involved pairwise correlations for each condition combination at each time point across EEG channels. The upper left corner shows the average TRF RDMs for both dialogue and background streams. The plot in the left corner displays the correlation between an attentional task model (attend speech vs. ignore speech, att. vs. ign.) and the 2 TRF RDM time series, with significant points displayed below the plot (FDR corrected, one-sample *t* test, *df* = 19). Right: Similar to TRFs, fMRI RDMs were constructed using searchlight SVM decoding across the 16 conditions, resulting in voxel-specific RDMs. Regions with above-average correlations between the attentional task model and fMRI RDMs are displayed (HPC parcellation). Shading indicates ± SEM. Code and processed EEG and fMRI data used to generate this figure are archived on the Open Science Framework; HTTPS://DOI.ORG/10.17605/OSF.IO/AGXTH. EEG, electroencephalography; RDM, representational dissimilarity matrix; SER, speech envelope reconstruction; SVM, support vector machine; TRF, temporal response function.

home/jkaurama/vbook/). This analysis indicated that attention facilitates recurrent feedforward-feedback loops in the ventral processing stream (see [2]).

## Results

### Attentional modulation of speech envelope reconstruction accuracy fluctuates during the course of the dialogue

We used the accuracy of SER to study how selective attention and our other experimental manipulations affected the neuronal tracking of AV cocktail party speech. We employed a 2 × 2 × 2 × 2 within-subjects factorial design, where participants performed 3 runs including

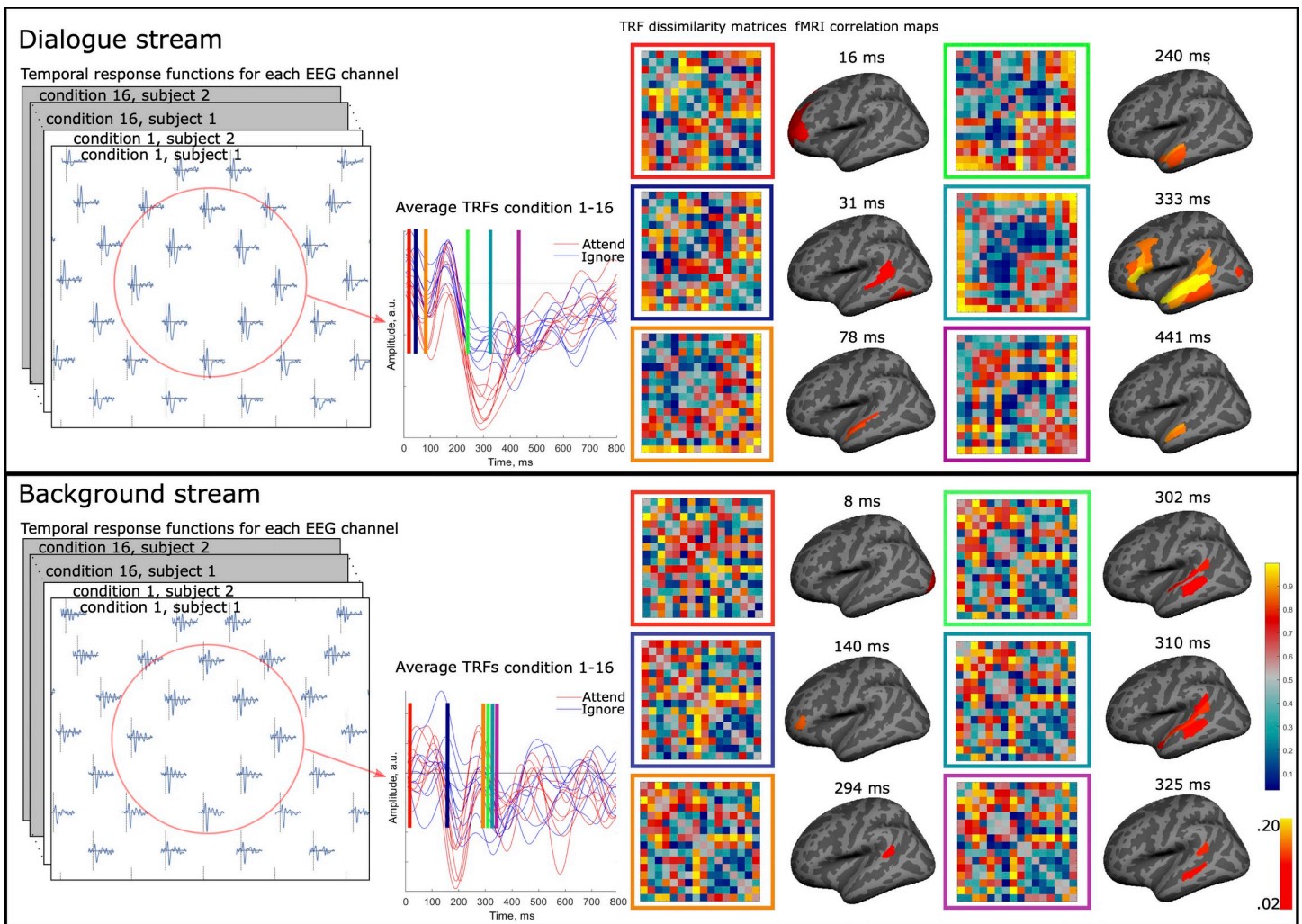

**Fig 4. Schematic illustration of TRF-fMRI fusion and results.** TRFs were separately estimated for the dialogue (upper part) and background streams (lower part) for each combination of semantic coherence and audiovisual quality and EEG channel. Average TRFs are displayed for frontocentral electrodes in the middle column (attend speech: red, ignore speech: blue). We constructed TRF RDMs for each time point by correlating each EEG channel TRF pairwise across conditions and participants. Similar fMRI RDMs were constructed based on SVM decoding between the 16 condition pairs from fMRI data. Thus, we constructed similar RDMs for the EEG and the fMRI, allowing us to fuse information from both datasets by correlating vectorised TRF RDMs with fMRI RDMs, controlling for task and opposite speech stream TRF RDMs (see Fig 3C). To identify fMRI activations which corresponded to TRF RDMs at different time points, we conducted one-sample $t$ tests ($df$ = 18, FDR corrected) averaged across the HCP parcellation ROIs. Six time points of this TRF-fMRI RSA analysis are displayed for both the dialogue (upper part) and background streams (lower part) on the right side of the figure. For the full-time series, refer to S1 and S2 Videos. Code and processed EEG and fMRI data used to generate this figure are archived on the Open Science Framework; HTTPS://DOI.ORG/10.17605/OSF.IO/AGXTH. EEG, electroencephalography; fMRI, functional magnetic resonance imaging; RDM, representational dissimilarity matrix; ROI, region-of-interest; RSA, representational similarity analysis; SVM, support vector machine; TRF, temporal response function.

all possible combinations of the Attentional Task (attend, ignore), Auditory Quality (good, poor), Visual Quality (good, poor), and Semantic Coherence (coherent, incoherent). To control for stimulus effects, each dialogue/background speech segment (across all conditions and runs) was unique (i.e., each was heard only once). The condition order and the dialogue allocated to each condition varied between participants (see "Procedure"). Because the dialogue stream comprised audiovisual speech, while the background speech comprised purely auditory speech, spoken by a speaker different from the ones having the dialogue, the main comparisons were conducted separately within speech streams.

In brief, multidimensional transfer functions were estimated based on all EEG channels for the dialogue streams and the background streams separately for each combination of Attentional Task, Semantic Coherence, Auditory Quality, Visual Quality, Line of the speech stream (1–7), and Segment of the line (1–4). Thereafter, the accuracy of the speech reconstruction was assessed by correlating the reconstruction with its corresponding speech envelope and correcting for spurious correlations (see "First-level analysis of EEG-data" and "Univariate analysis of EEG data" for details, and Fig 2A). The strength of the correlation between the SER and its corresponding speech envelope is generally assumed to reflect the accuracy of neuronal entrainment to the input speech [24]. We used linear mixed models to assess the effects of the repeated factors Attentional Task, Semantic Coherence, Auditory Quality, and Visual Quality on SER accuracy, separately for the dialogue stream and the background stream.

SER accuracy for the dialogue stream was significantly modulated by Attentional Task ($F_{1,18.7} = 67.2$, $p < 0.001$, $\eta^2 = 0.78$). That is, SER accuracy was higher in the task where participants attentively listened to the dialogue streams (mean $\Delta r = 0.14$, SEM = 0.004) than in the task where they ignored the dialogue stream (mean $\Delta r = 0.06$, SEM = 0.005). This is in line with previous studies that have shown that selective attention to a specific speech stream strongly increases neuronal tracking of that speech stream compared to the ignored speech streams [29,51,52]. Please refer to S1 Text and S1 and S2 Figs for all other significant effects in these linear mixed models and their correspondences to the behavioural performance results (e.g., effects related to Semantic Coherence, Auditory, and Visual Quality).

In our previous study utilising fMRI, we reported that in the auditory cortex attention-related modulations changed in a linear-quadratic fashion during the dialogue (i.e., increased in the beginning of the dialogue and abated thereafter; see Fig 2C, [38]). Neuronal tracking has been suggested to be most strongly affected by neuronal processes in the superior temporal cortex [29], and thus we expected similar temporal effects here. However, unlike with fMRI, here we could assess whether the previously reported temporal modulations were due to the processing of the attended or the ignored speech stream because they are separable in the EEG data. Further, utilising the EEG data in the present study also allowed us to evaluate whether attentional modulation changes within each line in a similar fashion as between the lines, which was not possible in our previous study due to the temporal resolution of fMRI.

Thus, first we examined within-line effects, i.e., whether SER accuracy changed within the line (lines were divided into 4 equal length segments). As seen in Fig 2B, there was a significant linear decrease in SER accuracy throughout the line for the dialogue stream when it was attended, and to some extent for the dialogue stream when it was ignored but not for any other combinations of Attentional Task and Speech Stream (Fig 2B; significant Condition × Segment interaction, $F_{9,116} = 11.2$, $p < 0.001$, $\eta^2 = 0.46$; linear mixed model with the repeated factors Condition (attend speech task dialogue stream; attend speech task background stream; ignore speech task dialogue stream; ignore speech task background stream) and Segment (1–4)).

Next, we analysed whether attention changed SER accuracy from line-to-line in a similar nonlinear fashion as previously seen in our fMRI study [38]. As seen in Fig 2D (left), SER accuracy showed a similar temporal profile changing from line-to-line as previously observed with fMRI. In other words, SER accuracy increased during the first lines of the dialogue and abated towards the end. Further, this temporal effect was only evident when the participants selectively attended to the dialogue stream (Fig 2D left; significant Condition × Line Number interaction, $F_{18,137.3} = 2.4$, $p < 0.002$, $\eta^2 = 0.24$; linear mixed model with the repeated factors Condition (attend speech task dialogue stream; attend speech task background stream; ignore speech task dialogue stream; ignore speech task background stream) and Line Number (1–7)).

Next, we considered the possibility that the slow temporal effects (i.e., line-to-line effects) we found in the SER accuracy data were only evident when analysing SER accuracies separately for each speech stream. That is, it might be that weaker neuronal tracking of the dialogue stream causes a similar concordant change in the neural tracking of the background stream, and thus the contrast between the 2 streams remained constant throughout the dialogues. Therefore, we performed a multivariate analysis that integrated information from both the dialogue stream and the background stream. Specifically, we assessed whether classification of trials as belonging to the attend speech task or the ignore speech task (using the SER correlations for the dialogue streams and the background streams as input) changed over the course of the dialogue (for details, see "Decoding analysis of SER accuracies"). This analysis revealed that decoding accuracy changed in a similar fashion as SER accuracies of the attended dialogue stream alone (Fig 2D, middle; linear mixed model with Line Number as the repeated factor and decoding accuracy as the outcome, $F_{6,25.6} = 2.7$, $p < 0.03$, $\eta2 = 0.39$).

Previous studies have shown that SER accuracy correlates positively with behavioural performance [29]. Therefore, attentional lability [53] during different parts of the dialogue could be considered a simple explanation for the slow temporal modulations. This attentional lability should also be observed in the behavioural performance. However, there was no significant change in the behavioural performance in the attend speech task across the lines of the dialogue (generalised linear model with Line Number as a repeated factor; $\chi^2_6 = 9.8$, $p > 0.13$, see also S2 Fig right, and [38]). Furthermore, unlike previous reports [29], we found no significant general association between SER accuracy in the attend speech task (dialogue stream) and behavioural performance. However, we found that the association between performance and SER accuracy changed over the lines of the dialogue (Fig 2D right, linear mixed model with Line Number as a repeated factor; $F_{6,\ 28.6} = 2.9$, $p < 0.02$, $\eta^2 = 0.37$). This temporal profile was inverse to the temporal profile of SER accuracy. That is, behavioural performance was negatively associated with the lines that showed the highest SER accuracy and positively associated with the lines that showed the lowest accuracy.

## Attention modulates temporal response functions of the attended and ignored speech streams

Speech reconstruction analysis has the advantage of maximising the power of finding effects in the EEG data, because it integrates information across channels and time points to estimate the optimal reconstruction of the sound stimulus. This, however, has the drawback of losing timing and location information in the neural signatures. Therefore, we also performed encoding modelling, separately for each combination of speech stream and listening condition (Fig 3A). In this model, the speech envelope was used as a regressor in a ridge regressor model, performed separately for data from each EEG channel (see "Univariate analysis of EEG data"). The output of this analysis is a TRF, which describes the convolution in time needed to translate the speech envelope into the EEG data. With some caveats, TRFs can be conceptualised as event-related potentials (ERPs) to a continuous variable, which here is the continuous speech amplitude envelope, and where the timescale refers to time lags to the speech signal (see "First-level analysis of EEG data") [11]. The caveats are that TRFs are filtered more heavily than standard ERPs (we used passband of 0.5 to 10 Hz) and the choice of regularisation smears exact temporal information, and due to this, the estimated timing of neural events cannot be assumed as exact as for standard ERPs.

We analysed whether selective attention modulates TRFs in frontocentral electrodes (optimal for picking up auditory cortex attention effects; e.g., [54,55]) separately for the dialogue stream (Fig 3B, left) and the background stream (Fig 3B, right). Selective attention significantly

enhanced the TRFs for the dialogue stream (i.e., there was a significant main effect of Attentional Task, i.e., attend speech > ignore speech) and this effect was present at 2 intervals between 0 and 800 ms, first at ca. 50 to 100 ms and then at ca. 200 to 400 ms after sound envelope changes (paired permutation $t$ tests, df = 18). This is consistent with previous ERP and TRF studies showing that attention modulates auditory processing of speech relatively early (i.e., within 100 ms), but that the strongest modulation is found at later time points [11,31,35,56–58].

Selective attention also changed both the timing and the amplitude of the TRF to the background stream (at ca. 50 to 200 ms; paired permutation $t$ tests, $df$ = 18). It is important to note that the background stream was ignored in all conditions. However, during the attend speech task, the participants had to actively suppress the background stream, while in the ignore speech task they focused on visual stimuli, designed to automatically keep attention away from all speech streams. Thus, since especially early components of the TRF response likely originate from the auditory cortex [35], it could be expected that the early components of the TRF to the background stream would be smaller for the attend speech task than the ignore speech task. However, our results indicated the reverse. This pattern might arise if participants had involuntary momentary lapses of attention to the wrong speech stream [11] during the attend speech task, causing enhancements also in the background speech stream. We find this unlikely, however, because such lapses would likely cause more variance in the background stream TRFs, rather than the change in amplitude seen in the present results. Furthermore, previous studies using the same paradigm [42] have found that in general, participants do not remember topics of the background stream.

## EEG TRF–fMRI fusion reveals that attention facilitates several feedforward-feedback loops related to the processing of cocktail-party speech

Next, we performed multivariate RSA on the TRFs. TRFs were estimated for each condition (16 conditions, 8 attend speech task, 8 ignore speech task, for the exact order of the conditions, see "Multivariate analysis of TRFs") and channel (128 channels) separately for the dialogue and the background streams. Thereafter, for each sample of the TRFs (128 Hz, ca. 8 ms samples), we performed pairwise correlations across the EEG channels for each condition pair to construct dissimilarity matrices (*1-r*). This resulted in 1 TRF representational dissimilarity matrix (RDM) for each time point of each speech stream. An RDM is a geometrical description of the data, showing an assembly of all pairwise dissimilarities across neural responses or model predictions to different stimuli or experimental conditions [59]. As can be seen in Fig 3C (upper left corner), especially in the dialogue stream, the attend speech task conditions are generally similar to each other and dissimilar to the ignore speech task conditions, i.e., there is an effect of Attentional Task. To test when this effect was significant, we constructed a model matrix for the main effect of Attentional Task (Fig 3C, upper left corner) and correlated this model with the TRF RDMs for each time point (one-sample $t$ test, FDR corrected across time points; for other model correlations, see S3 Fig). This analysis revealed that selective attention modulated TRFs throughout almost the whole time range from 100 ms (after the sound envelope changes) onwards (Fig 3C, lower left corner). For the background stream, there were significant correlations with the attentional task model between 150 and 300 ms.

We also performed the same RSA analysis on our fMRI data that used the same paradigm but different participants (the same fMRI data were used for the TRF-fMRI fusion, see below). Here, dissimilarity matrices were generated based on pairwise searchlight SVM decoding between the 16 conditions (see "Decoding analysis on the fMRI data" for details). Fig 3C (lower right) shows the regions (averaged for each region of the human connectome project

atlas, HCP parcellation [60]), where the correlation with the attentional task model was above average (i.e., $r > 0.34$, the threshold for significance was $r > 0.08$). This analysis shows that information that distinguishes the attend speech task from the ignore speech is contained globally in the brain (see also, [38]) which also probably partly explains why the attentional task model correlated with TRF RDM matrices throughout the time interval.

To gain an understanding of how the TRF RDMs corresponded to the fMRI RDMs, we performed EEG-fMRI fusion [45,46], using TRF RDMs, estimated using the EEG data, and fMRI RDMs (see section "TRF-fMRI fusion" for details). This was achieved by correlating each TRF RDM with the fMRI RDMs averaged in each region-of-interest (ROI) from the HPC parcellation. Because the fMRI RDMs integrate the differences for both the dialogue and background stream, while the TRF RDMs separate these effects, we corrected the TRF-fMRI fusion for the TRF RDMs of the opposite speech stream. That is, the dialogue stream TRF-fMRI fusion was corrected for the background stream TRF RDMs and vice versa. We also corrected for the main effect of Attentional Task in the TRF-fMRI fusion analysis because this effect was global in both the TRF and the fMRI responses (see above; Fig 3B and 3C), and thus masks subtle differences between different regions. Thus, the main effect of Attentional Task (Fig 3B and 3C) does not contribute to the TRF-fMRI fusion, and the correspondences arise instead due to other more complicated correspondences between the EEG and fMRI RDMs (see below).

As seen in Fig 4 (upper right corner) and S1 Video (www.mv.helsinki.fi/home/jkaurama/vdialog/) for the dialogue stream, the first significant correlations (one-sample $t$ test, $df = 18$, FDR-corrected) between the TRFs and fMRI RDMs arose at ca. 16 ms after sound envelope changes in dorsolateral and dorsomedial frontal regions. Hereafter, correlations arose at ca. 30 ms in posterior auditory regions and slowly thereafter in anterior auditory cortical regions. After 150 ms correlations arose in the anterior temporal lobe and slowly spread (ca. 250 ms) back to auditory, frontal, and speech processing regions in an anterior–posterior fashion. A second anterior–posterior sweep in the auditory cortex occurred starting at ca. 450 ms. Note that the timing information in this analysis is derived entirely from the TRF data and the spatial information from the fMRI data. For the background stream (Fig 4, bottom right corner, S2 Video, www.mv.helsinki.fi/home/jkaurama/vbook/), there were initially correlations in the visual cortex. At ca. 140 ms, there were correlations in the dorsolateral frontal cortex and then at ca. 300 ms correlations arose in the auditory cortex moving in a posterior–anterior fashion. This effect may relate to suppression of the background stream (cf. [61]). The TRF-fMRI correlation patterns for the background stream were lateralised to the left, while for the dialogue stream they were bilateral, which is consistent with earlier neuroimaging studies [62].

It is important to note that these TRF-fMRI fusion patterns were not due to a main effect of Attentional Task (Att. versus ign.) since this effect was controlled for in the analysis. Furthermore, as can be seen in S3 Fig, no other main effect model or interaction model yielded FDR-corrected significant results. However, based on the uncorrected results (S3 Fig), it seems that the correlations were mostly influenced by interactions between Attentional Task (Att. versus ign.) and stimulus features. However, many of the correlation patterns in the TRF-fMRI fusion analysis likely arose due to idiosyncratic differences between the different task conditions at different time points of speech processing. For time series in different a priori regions of interest, please see S4 Fig.

## Discussion

Our SER analyses on the EEG data replicated that selective attention enhances neural tracking of attended speech [25,29,30]. Similarly, we replicated that attending to a specific speech stream enhances its EEG TRFs, both at early latencies (ca. 30–150 ms, e.g., [31]) and later

latencies (ca. 200–400 ms, e.g., [35]). These findings are consistent with the view that selective attention increases the contrast between attended speech and distracting speech through top-down neural signals, which propagate from higher-level cortical regions to sensory regions and serve to enhance the gain of neurons that process the relevant speech [12–14]. While this is a likely explanation for some of our observations, we find it highly unlikely that this model exhaustively explains how attention modulates sensory processing in the auditory cortex, which we will discuss below.

Although the background stream was always ignored, TRFs for the background stream were both temporally expedited and amplified when participants listened to the dialogue stream compared to when both streams were ignored. Thus, it seems that selective attention not only enhances the processing of relevant speech but also modulates the processing of the actively ignored distracting speech (for similar findings, see [11,63]). Such modulations might reflect active suppression of auditory cortex neurons processing attributes of distracting speech, which has been suggested as a complementary mechanism to increase the contrast between attended sounds and ignored sounds [56,63–65]. Alternatively, the effects may reflect that early processing of the background stream cannot be suppressed when attending to speech [32], attention enhancements partially spread to the background stream [66] or attention fluctuates between the 2 streams [11]. As the background speech is affected by manipulations of the attended speech, future studies could do manipulations for both sound streams (dialogue and background streams) and alternate the focus of attention between the 2 streams. Later studies utilising source localisation and/or intracranial measurements could also reveal both the spatial and laminar attributes of these effects and the neural populations contributing to them.

RSA of the EEG data for both the dialogue stream and the background stream revealed that selective attention strongly modulated TRFs at several latencies that have not been reported in previous studies. Note that unlike the univariate TRF analyses the RSA analyses utilised all EEG-channels and were thus expected to find significant patterns in the channels outside those usually studied in auditory attention studies (e.g., frontocentral electrodes). Corroborating this, RSA analyses on the fMRI data showed that attention modulated information processing in extensive cortical fields not limited to regions associated with speech processing or executive functions (see also [63]). Thus, these results cast doubt on models that highlight simple interactions between frontal and sensory neural networks as origins for selective attentional effects. Rather, our results suggest that selective attention modulates a multitude of different subprocesses widely distributed in the brain (see also [38]).

Using RSA, we performed TRF-fMRI fusion, which showed that attentional modulation of information flow between sensory regions and higher-level regions displayed reliable spatial and temporal characteristics. The earliest modulations were found in the lateral, medial, and inferior frontal cortices at around 8 to 16 ms. This is consistent with earlier MEG source localisation of attentional effects on speech related TRFs [11] and might reflect preparatory signals biasing the attentional speech processing (e.g., when the quality of the sensory input is poor). Thereafter, information flow generally followed the ventral stream model [2], with information processing first being modulated in the secondary auditory cortex (around 30 ms), continuing anteriorly to the superior temporal cortex and finally to the anterior temporal lobe (at around 150 ms). At later latencies (after 200 ms), several back-propagating loops of information flow between the anterior temporal cortex, frontal cortex, and the auditory cortex can be discerned. This suggest that information flow during active processing of cocktail-party speech is associated with reverberant bidirectional (feedforward-feedback) informational flow from sensory regions to regions associated with semantic [8], syntactic [9], and executive functions [67], within the ventral processing stream.

As previously mentioned, we found that attention enhanced the neural tracking of the attended speech. However, this modulation was not uniform in time, i.e., the SER accuracy linearly decreased within the line of the dialogue (ca. 5 s long). This type of decrease could be explained within a predictive coding framework [68], assuming that information accumulates as the line proceeds, which constrains prediction error (PE) in neural networks [51,69]. Importantly, however, we found that decreases in SER were most consistently observed for attended speech. Thus, if the SER temporal profile is explained by predictive mechanisms, such mechanisms seem to depend on selective attention. Indeed, some current models postulate that predictive coding mechanisms and selective auditory attention interact during attentive processing of sensory information (cf. [19,70,71]). In the model proposed by Schröger and colleagues [19], the attentional processing of relevant sounds is biased in the auditory cortex through recurrent loops, with higher-order processing networks establishing an "attentional trace" which maximally distinguishes the features of the attended sounds from the features of the irrelevant sounds. In this model, selective attention improves the precision and gain of PEs generated by neurons encoding the attended stimuli. These enhanced error signals are concurrently sent to regions at the higher level of the processing hierarchy, which in turn send stronger modulatory signals to lower levels of the hierarchy. Thus, attention may influence feedback/feedforward loops, which interact with, for example, the predictability of the input. This model seems to explain quite well the present linear decrease effects. The model also gives a framework for understanding our TRF-fMRI fusion results, suggesting that the recurrent feedforward/feedback loops reflect the propagation of PE from the lower level of the hierarchy to the next level, on the one hand, and correcting predictive signals from the higher level to the lower level, on the other.

We also found that the strength by which selective attention enhances neural tracking of speech changes on a slow temporal scale (from line-to-line of the dialogue, Fig 2C). In contrast to the linear decrease seen within a line, the neural tracking first increased up to the middle of the dialogue, and thereafter decreased towards the end of the dialogue. Similar slow fluctuations of attentional effects have been previously described using fMRI [38,39] and behavioural experiments (e.g., [72]). From the predictive coding framework, it could be postulated that such a temporal profile would arise if the ability of attention to maximally increase the gain of PEs takes time to build up, causing an initial increase in SER. The subsequent decrease could be explained, as for the within-line effects, by predictions becoming more stable towards the end of the dialogue. This account, however, fails to explain why there is no indication of such a delay in facilitating attentional processes within the line. Furthermore, based on this account, behavioural performance would be expected to improve as the dialogue proceeds and the model of the heard speech becomes increasingly accurate. We did not, however, find any evidence for such changes in the behavioural performance data. Importantly, in our previous publication on the fMRI data, we reported similar slow temporal changes of attention-related modulations in the superior temporal cortex [38]. In that paper, we suggested that the temporal modulations arose due to recruitment of additional neuronal resources in speech networks that may aid in automatising speech processing. This account is based on the model proposed by Kilgard [20], originally used to explain why attention and plasticity initially recruit neurons in the sensory cortex, which after task automatisation no longer participate in the task. Several animal studies have shown that attentional tasks cause transient–persistent plastic changes in auditory neuronal response profiles (e.g., [73,74]). The conundrum, however, has been that some studies have indicated that behavioural performance accuracy persist after the original plastic changes have subsided [75]. Therefore, Kilgard proposed that when the task is initially learned, all possible neuronal networks that may be useful to solve the task at hand are recruited. Gradually, the unnecessary, less informative neuronal networks are pruned out, and

the most efficient network ends up performing the task (sparse coding). Thus, the slow temporal profiles seen in the current study may reflect that in the sensory cortex, neurons that may help in building the attentional trace are initially recruited and subsequently pruned out to encode information in a maximally sparse manner. This account would also explain the present perplexing performance-SER association (Fig 2C). That is, we found that behavioural performance predicted SER accuracy negatively in the middle of the dialogue when SER accuracy was strongest and positively when accuracy was weakest. Thus, it may be that behavioural associations were negative in the middle of the dialogue because at this point, neuronal resources processing the speech may not necessarily help in performing the task, while towards the end of the dialogue, unnecessary units are pruned out and the association between SER and performance returns to positive.

Models of the auditory system have generally overlooked how factors like attention and active tasks influence the processing of sounds in neural networks. This oversight relies on the premise that attention simply changes neuronal response gain. Our results, however, highlight that the enhanced neuronal tracking of attended speech is not necessarily uniformly associated with more accurate representation of the attended speech (see e.g., [29]) but changes as a function of time due to predictive and/or other nonlinear plastic mechanisms in sensory cortex. We argue that the approach to selective attention needs to be updated to reflect recent views on how cognition is organised in neural systems (see e.g., [76]). Instead of mechanistic models where higher-level networks enhance gain mechanisms in sensory neurons, attention could be modelled as a collection of temporally changing processes that route activity in distributed neural networks according to behavioural demands. These findings may offer key insights in improving dynamic computational models of selective attention in noisy conversational settings (see e.g., [77]). Current AI platforms struggle to match human listeners and deliver unsatisfactory performance. Later multi- and single unit recordings in the auditory cortex could test the hypothesis that attention both changes the gain of neuronal populations and initially recruit neuronal resources that may aid in the performance of the task that are later discarded due to optimisation of task performance.

## Methods

### Experimental model and study participant details

**Participants.**  EEG data were collected from 20 adult university students at the University of Helsinki and Aalto University (11 females, age range 19 to 28 years, mean 23.4 years). One participant was excluded due to a technical problem with the EEG data acquisition. fMRI data were collected from a separate sample of adult university students at the University of Helsinki and Aalto University comprising 23 adult participants (14 females, age range 19 to 30 years, mean 24.3 years). fMRI data were excluded based on preestablished criteria. Two participants were excluded due to excessive head motion ($>5$ mm) and 2 participants due to anatomical anomalies that affected coregistration. Thus, data from 19 participants were used in the analyses. The fMRI data has been previously analysed and published in [38] but in the present manuscript, the data were analysed differently, yielding previously unreported results, e.g., fusion with the EEG-data. All participants were monolingual native Finnish speakers, and they did not have any self-reported neurological or psychiatric diseases. In addition, they had self-reported normal hearing and normal or corrected-to-normal vision. All participants were right-handed, and this was confirmed by the Edinburgh Handedness Inventory [78].

**Ethics statement.**  The studies involving human participants were reviewed and approved by Ethics Review Board in the Humanities and Social and Behavioral Sciences, University of Helsinki (number: 14/2017). The research follows the ethical guidelines of the Declaration of

Helsinki. The participants provided their written informed consent to participate in this study. Written informed consent was obtained for the sharing of processed anomynised data from each participant. The 2 people visible in Fig 1 and the photographer gave written consent for the publication of the identifiable images under the Creative Commons By 4.0. license.

## Method details

**Preparation of stimulus materials.** The stimuli comprised dialogues between 2 (female and male) native Finnish speakers. Written informed consent has been obtained from the individual(s) for the publication of any potentially identifiable images or data included in this manuscript (see also [38–40,42]). The dialogue topics were about neutral everyday subjects such as the weather. The dialogues comprised 7 lines (ca. 5.4 s of duration) followed by a ca. 3 s break (2.9 to 4.3 s), resulting in a total length of 55 to 65 s (mean 59.2 s) for each dialogue. The speakers spoke their lines in an alternating fashion; the female talker started the conversation in half of the video clips.

The original dialogues [42] were recorded so that the talkers sat next to one another with their faces slightly tilted towards each other (see Fig 1A). For more details on the recordings, see [42].

In both the EEG and the fMRI experiment, we used 24 of the original dialogues for the coherent context conditions. The rest of the dialogues were used to construct 24 new dialogues for the incoherent context conditions. These semantically incoherent dialogues were constructed by shuffling lines from different dialogues of the 36 original dialogues. Dialogues were chosen based on the location and posture of the speakers so that there would be minimal visual transition between each line of the shuffled dialogues. Because slight differences in lighting and posture of the speakers, we divided the videos into pools of 6 videos that were maximally similar. In the semantically incoherent dialogues, each of the 5 lines were from a separate dialogue, and the remaining 2 from one dialogue. To secure that all lines were equally unpredictable, we ensured that the 2 lines from the same original dialogue were separated by at least 4 other lines.

The semantically incoherent dialogues were constructed by first removing the audio stream from the video, whereafter the video image was edited with Adobe Premiere Pro CC software with the morph-cut function (Adobe, San Jose, California, United States of America). To prevent participants from noticing these changes, the transition from one dialogue to another always occurred on the side where the talker was silent (see [38], Supplementary video material 1–8; https://osf.io/agxth/). The lighting was edited to fade small differences between the different clips.

Two small grey squares (size 1.5˚ × 1.5˚) were added to the videos below the faces of the speakers. A white cross (height 0.5˚) was placed in the middle of the square below the face of the talker who was speaking at that given moment. This cross faded out immediately as the talker ended their line and reappeared 1.5 s later. Thus, most of the time, there were 2 crosses present in the video (see [38], Supplementary Video material 1–8; unlike in our experiments, these videos have English subtitles). In the visual control task, the disappearance of the cross indicated that the participant should turn their attention to the other side of the video frame. The cross changed from a plus sign (+) to multiplication sign (×) or vice versa, randomly 9 to 15 times during each dialogue. The cross rotated only on the side where the talker of the dialogue was speaking. During each of the 7 lines, the cross rotated 1 to 4 times, i.e., every 1.25 to 2.5 s.

The audio streams were noise-vocoded before adding the audio streams back to the videos [42]. This was achieved by dividing the audio streams into 4 (poor auditory conditions) and 16 (good auditory conditions) logarithmically spaced frequency bands between 0.3 and 5 kHz

using Praat software [version 6.0.27, 47]. The talkers' F0 (frequencies 0 to 0.3 kHz) was unchanged (see [42] for details).

To manipulate the amount of visual speech seen by the participants, we added a dynamic white noise masker onto the speakers' faces (see [42]).

Finally, the poor and good quality audio files were recombined with the poor and good visual quality videos with a custom Matlab script.

As the final step, we added a continuous background stream to the dialogues. We used a freely available audiobook about cultural history (a Finnish translation of *The Autumn of the Middle Ages* by Johan Huizinga, distributed online by YLE, the Finnish Broadcasting company), read by a female native Finnish professional actor. The F0 of the reader was lowered to 0.16 kHz and the audiobook was low-pass filtered at 5.0 kHz [42].

**Procedure.**   The videos, including the dialogue stream and background stream described above, were used in our 16 experimental conditions defined by Attentional Task (attend speech, ignore speech; Att. versus ign.), Semantic Coherence (coherent, incoherent), Auditory Quality (good, poor), and Visual Quality (good, poor). We presented 3 runs, each containing 8 of the 24 coherent video clips (in all coherent context conditions) and 8 of the 24 incoherent video clips (in all incoherent context conditions). Thus, all the participants were presented with all the 48 dialogues. Every other run started with the attend speech task, and every other with the ignore speech task. Within the functional runs, the attend speech task and the ignore speech task were presented in an alternating order. The order of conditions and dialogues presented was pseudorandomised. Because we could not entirely randomise the videos into the 16 conditions per run, we used the Latin square to construct 4 different versions of the experiment (see Suppl. Table 3 in [38]).

Stimulus presentation was controlled by using Presentation 20.0–22.0 software (Neurobehavioral Systems, Berkeley, California, USA). The auditory stimuli were presented binaurally through insert earphones (Sensimetrics model S14; Sensimetrics, Malden, Massachusetts, USA). Before the experiment, the audio volume was set to a comfortable level individually for each participant. It was approximately 75 to 86 dB SPL at the ear drum. During EEG, the video clips (size 26˚ × 15˚) were presented in the middle of a 24-inch LCD monitor (HP Compaq LA2405x; HP, Palo Alto, California, USA) that was at ca. 40 cm from the eyes of the participant. During fMRI, the video clips (size 26˚ × 15˚) were projected onto a mirror attached to the head coil and presented in the middle of the screen. Videos were presented on a uniform grey background. In the middle of each run, there was a break of 40 s. During the break, the participants were asked to rest and focus on a fixation cross (located in the middle of the screen, height 0.5˚). The distracting audiobook (presented with a sound intensity 3 dB lower than the voices of the viewed male and female speakers) started randomly 0.5 to 2 s before video onset and stopped at the offset of the video. The differences in dialogue durations were compensated by inserting periods with a fixation cross between the instruction and the onset of the dialogue, keeping the overall trial durations constant.

**Tasks.**   During the attend speech task, the participants were asked to attend to the 2 speakers having a discussion in the videos while ignoring the background speech. After every dialogue, the participants were presented with 7 statements relating to the occurrence of a topic in each line from the dialogue by pressing the "Yes" or "No" button on a response pad with their right index or middle finger. Questions were for example, "Did the boy drop his phone?", "Was there a cat on the table?". A new statement was presented every 2 s. After the 7 statements, the participants were provided with feedback on their performance (number of correct responses).

During the ignore speech task, the participants were asked to attend to the fixation cross presented in the videos and calculate how many times the cross rotated from a multiplication

sign (×) to a plus sign (+) and vice versa. Every time the cross disappeared, the participants were supposed to shift their attention to the other fixation cross on the other side of the frame. The participants were instructed to actively ignore all speech stimuli, i.e., the dialogues and the audiobook. At the end of the video, the participants were presented with 7 statements about the rotating cross ("Did the cross turn X times?", the X being between 9 and 15 in an ascending order). As in the attend speech task, the response was given by pressing either the "Yes" or "No" button on a response pad. If the participants were unsure, they were instructed to answer "Yes" to all the alternatives they deemed possible. After the 7 statements, the participants received feedback on their performance (number of correct responses).

**Additional task.**   After completing the 3 runs, the participants were presented with an additional run consisting of a single dialogue and one set of 7 questions (note only in the EEG experiment). The dialogue employed in this extra run was the one of the 12 original coherent dialogues that were used to create the 24 incoherent dialogues (i.e., these 12 dialogues had not been seen/heard by the participants in their coherent form in the present experiment). The purpose of this additional run was to evaluate how much the participants processed the semantics of the dialogues they were instructed to ignore during the visual control task. The participants were presented with a dialogue video, and they were told to complete the visual control task and hence ignore the dialogue while counting fixation cross rotations. At the end of the video, they were, however, instructed to answer 7 yes-no questions about the lines of the dialogue. The dialogues in this additional run were presented with good auditory and visual qualities and with a coherent semantic context as this was considered the type of conversation that would be the hardest to ignore. This task concluded the experiment. Thus, the additional task was completed by 19 participants participating in the EEG experiment. For results on this task, please see [38].

**Pre-trial.**   Before the experiment, all participants practised the tasks. In the practice phase, the participants performed the attend speech task and the ignore speech task, using a coherent dialogue not included in the actual experiment. The dialogue was presented with different auditory and visual qualities.

**Data acquisition.**   The EEG data were collected at the Department of Psychology and Logopedics, University of Helsinki, in a soundproof and electrically shielded EEG laboratory. The data were registered separately for each of the 3 runs of each participant, and the overall duration of the EEG measurements was approximately 1.5 h per participant. The EEG data were recorded with a BrainVision actiCHamp amplifier (128 channels) and a BrainVision acti-CAP snap electrode cap with an actiCAP slim electrode set of 128 active electrodes (Brain Products GmbH, Gilching, Germany). The electrode layout was an extended version of the International 10–20 system, and recording reference was at FCz. The amplifier bandwidth was 0 to 140 Hz and the sampling rate was 500 Hz. The EEG data were recorded with BrainVision Recorder (version 1.21.0402–1.22.0002; Brain Products GmbH, Gilching, Germany). Electrode impedances were checked prior to recording, and they were below 10 kΩ for most electrodes for most participants. When needed, worsened impedances were enhanced in-between the experimental runs.

For a detailed description of the fMRI acquisition, see [38]. We report the parameters used in brief in Table 1.

## Quantification and statistical analysis

**Analysis of behavioural data.**   The total number of questions in the experiment was 336 (48 dialogues × 7 lines). We registered the number of correct answers in each task block. Misses were treated as incorrect button presses. The mean task performance and standard

**Table 1. MRI-acquisition parameters used in the fMRI data collection (see [38] for details).**

|     | TR | TE | Flip angle | Voxel matrix | Slice thickness | FOV | Slice | Resolution |
|-----|-----|-----|-----|-----|-----|-----|-----|-----|
| **EPI** | 2.6 s | 30 ms | 75˚ | 64 × 64 | 3.0 mm | 19.2 cm | 43 | 3 × 3 × 3 mm |
| **T1** | 2.5 s | 3.3 ms | - | 256 × 256 | - | - | - | 1 × 1 × 1 mm |

error of mean were used to establish that the participants were performing the task as expected. To analyse participants' performance (EEG/fMRI experiment) during the attend speech and ignore speech task, 2 separate repeated-measures analyses of variance (ANOVA) were computed with 3 factors: Semantic Coherence (coherent, incoherent), Auditory Quality (good, poor), and Visual Quality (good, poor). ANOVAs were chosen instead of linear mixed models for these analyses to yield comparable results to those reported for the fMRI experiment, which are not reported in the present manuscript, but can be found in [38].

We also analysed the performance line-by-line to evaluate whether participants' performance changed during each dialogue (only performed for the attend speech condition performance data gathered in the EEG experiment). Here, we used a generalised linear model (identity link function) with the participant added as a random effect (including intercept) and the effect of line was modelled as a categorical repeated measure. The model was run using maximum likelihood estimation with a maximum of 100 iterations to converge.

Statistical analyses were carried out with IBM 18 SPSS Statistics 25 (IBM SPSS, Armonk, New York, USA) software and the results were visualised with Python (Mathworks, Natick, Massachusetts, USA).

**Preprocessing of EEG data.** EEG data preprocessing was carried out using the MNE Python 0.22 [79]. All channels were referenced to an average reference. Next, the data were manually inspected for channels that would subsequently be interpolated. At least one of the following criteria had to be met for a channel to be chosen for interpolation, and the criterion had to be present persistently throughout at least one of the 3 experimental runs. The criteria were a flat line response, high-frequency deviation, electrode pop artifacts, and body movement artifacts. The deviant channels were temporarily removed from the data.

An independent component analysis (ICA) was fitted on the concatenated runs for each participant separately, using MNE-ICA (picard-type). For each participant, we defined 2 to 4 components to be denoised that were classified as either blinks, lateral eye movements, or heartbeats. Thereafter, the raw data from each run was denoised using MNE.ica.apply. After this, the formerly chosen deviant channels were interpolated, the data was bandpass filtered (0.5 to 10 Hz) using the mne.raw.filter function, with the "firwin" option (default settings). This function computes the coefficients of a finite impulse response filter (hamming window). Thereafter, the data were down sampled to 128 Hz for the TRF analyses and 64 Hz for the speech reconstruction analyses. Thereafter, the EEG time series were cut into 6.5 s epochs based on the dialogue speech trials (see below).

**First-level analysis of EEG data.** To estimate the neural response to the 2 speech streams (dialogue stream and background stream), we performed speech tracking, using both an encoding and a decoding approach. In the encoding approach, we estimated TRFs for each EEG channel. In the decoding approach, we reconstructed the speech using data pooled across all 128 EEG channels (SER analysis).

The rationale for TRF estimation has been described in detail elsewhere (see e.g., [80]). In brief, TRFs constitute linear transfer functions describing the relationship between features of the stimulus function (S) and the response function (R; i.e., the EEG channel data). Stimulus features were constructed by extracting sound amplitude envelopes separately for the dialogue stream and the background stream using a Hilbert transform. The envelopes were band-pass

filtered (0 to 10 Hz) and down sampled to 128 Hz for TRFs and 64 Hz for SERs (a lower sampling rate was chosen to speed up analysis for SERs). Thereafter, the envelopes were cut into separate lines (6.5 s) for both sound streams.

In the encoding approach, 2 separate TRFs were estimated per EEG channel (dialogue and background; Fig 2A). These TRFs can be conceptualised as a linear composition of partially overlapping neural responses at different time lags ($\tau$) to a continuous stimulus, and they are therefore conceptually similar to ERPs ([11]). We estimated TRFs with the receptive field function (MNE-python: based on the mTRF toolbox that utilises ridge regression), with time lags −200 to 800 ms, and a common regularisation parameter ($\lambda$) of $10^5$ ([80]; see Fig 2A). Note that the regularisation parameter used affects the shape and amplitude of the TRF curves (for simulations, see e.g., [11]) and therefore we chose a common regularisation parameter (based on [80]) and used it in all conditions and participants.

In the decoding analysis, a multidimensional transfer function was estimated using all EEG channels as input (R) in an attempt to reconstruct separately the dialogue stream and the background stream amplitude modulations (see Fig 3A) using the receptive field function (MNE-python), with time lags ($\tau$) of −200 to 0 ms and a common regularisation parameter ($\lambda$) of $10^4$ [80]. Unlike the encoding analysis, this analysis yields stimulus construction for each time point of the stimulus function (see Fig 3A).

Both models used a leave-one-out approach, where in each iteration all trials (except one) are selected to train the model (train set), which was then used to predict either the neural response at each EEG channel (TRF) or the speech envelope of speech streams (SER) in the left-out trial (test set). This procedure was repeated with a different train - test partition in each iteration averaged over all iterations.

**Univariate analysis of EEG data.**   For the TRFs, we tested whether Attentional Task (i.e., attend speech task versus ignore speech task) modulated the TRFs averaged across 7 fronto-central electrodes (Cz, FCC1h, FCC2h, FC1, FC2, FFC1h, and FFC2h; Fig 3A), separately for the dialogue stream and the background stream for each time bin employing permutation paired *t* tests (20,000 permutations) using custom scripts written in Python.

For SER, in accordance with [29] we calculated correlations (Pearson) between the original speech stream envelopes and their reconstructions. Thereafter, we cross-reconstruction correlated the stimulus reconstructions and the stimulus envelopes (e.g., correlation between the dialogue speech amplitude and the background speech reconstruction which should be close to zero (Fig 3A)). Finally, to estimate SER accuracy we used correlation difference scores ($\Delta r$; between the correct reconstruction correlations and the cross-reconstruction correlations (Fig 3B)). For segment-level analysis, we divided the stimulus envelopes and the stimulus reconstructions into 4 segments of equal length and calculated correlations based on these instead of the full line.

The SER accuracies were analysed with different linear mixed models using IBM 18 SPSS Statistics 25. All models included the participant as a random effect and intercept. For repeated factors, the diagonal covariance structure was chosen. If there was more than one repeated factor, a random slope was added for all repeated main effects and interactions using the variance components method. The models were estimated using restricted maximum likelihood estimation with a maximum iteration number of 100 to converge and *df*-estimation was performed using Satterthwaite. Because SPSS does not produce effect size estimates for the fixed effects in linear mixed models, we used the formula partial $\eta^2$ = F × df$_1$ / (F × df$_1$ + df$_2$) [81] to approximate effect sizes where applicable.

To analyse how performance in individual trials affected the dialogue stream reconstruction during the attend speech conditions, we performed a similar two-level analysis commonly used when analysing fMRI data [38]. First, we created separately for each participant a linear

regression model with response (correct or incorrect) in each trial as the predictor and SER accuracy for that trial as output. Because there were not enough incorrect responses in any one sub-condition (e.g., coherent, good auditory, and good visual quality), all trials were pooled across the 8 stimulus conditions. However, because the quality manipulations might affect performance—SER accuracy associations, we added Semantic Coherence, Auditory Quality, and Visual Quality as confounds in this model. The β-weight for response was thereafter taken to the second-level analysis. The second-level analysis was a similar linear mixed model as described above with line entered as the repeated predictor and 5% trimming used for the output variable to remove noise due to the paucity of incorrect trials.

**Decoding analysis on the fMRI data.** The preprocessing and first-level analysis pipeline for the fMRI data were the same as that described in detail in [38] (for a brief description, see Table 2).

Support vector machine (SVM) decoding with leave-one-run-out cross-validation [82] was used to classify each pair of the 16 conditions (Task (attend speech task, ignore speech task) × Semantic Coherence (coherent, incoherent) × Auditory Quality (good, poor) × Visual Quality (good, poor)) in the fMRI data. Each line constituted an exemplar and each voxel a feature in the analysis. The SVM was conducted with the decoding toolbox (TDT, [83]) using the beta images from the first-level GLM in the participants' anatomical space. We used searchlight-based decoding [84] with a radius of 6 mm (isotropic), and with default settings of TDT; L2-norm SVM with regularising parameter C = 1 running in LIBSVM [85]. The resulting accuracy maps for each condition pair were thereafter projected to the Freesurfer average (fsaverage) using the participants' own Freesurfer surface (surface smoothing: 5 mm$^2$ full-width half maximum smoothing). The pairwise decoding accuracies were averaged within each 360 ROIs (HCP parcellation [60]) and RDMs ([86]) were constructed for each subject and each ROI. All RDMs in this study are displayed rank-scaled and the conditions are ordered so that the 8 attend speech task conditions are first and the ignore speech task are second. The coherence and quality conditions are in the following order (coherent: co, incoherent: inco, good: g, poor: p, visual quality: v, auditory quality: a; co-gv-ga, inco-gv-ga, co-pv-ga, inco-pv-ga, co-gv-pa, inco-gv-pa, co-pv-pa, inco-pv-pa).

The fMRI RDMs were compared with the attentional task model (Fig 3C). First, model and data RDMs were vectorised (lower triangular) and then correlated (Spearman $r$) with each other for each ROI and each participant. The statistical significance of the mean correlation above zero was tested with right-tailed $t$ test, FDR-corrected for 360 ROIs.

**Decoding analysis of SER accuracies.** SVM decoding with leave-one-run-out cross-validation [82] was used to classify SER correlations as either belonging to the attend speech task or the ignore speech task. Each line constituted an exemplar and the 4 correlations used to define SER accuracies in the univariate analyses (see above) were used as features ($r$: reconstruction of the dialogue stream envelope × the dialogue stream envelope, reconstruction of the dialogue stream envelope × the background stream envelope, reconstruction of the background stream envelope × the background stream envelope, reconstruction of the background stream envelope × the dialogue stream envelope). The SVM was conducted with the decoding toolbox (TDT, [84]) using the SER correlations from each participant, with default settings of

**Table 2. fMRI preprocessing parameters and first-level GLM specifications (see [38] for details).**

| Preprocessing | Motion correction (SPM) | Slice timing correction (SPM) | High pass filter (130 Hz) (SPM) | Pre-withening (SPM) | Normalisation to anatomical (BBR; FSL) |
|---|---|---|---|---|---|
| First-level analysis (SPM) | 112 EVs (16 × 7 lines) | 6 Motion correction nuisance EVs | Instruction | Quizzes | Canonical HRF convolution |

TDT; L2-norm SVM with regularising parameter C = 1 running in LIBSVM [85], 100 itera-tions. The resulting accuracies were thereafter analysed using linear mixed models.

**Multivariate analysis of TRFs.** RDMs were constructed from the dialogue TRFs as well as background speech TRFs separately for each time point by calculating 1-$r$ (Spearman) of all conditions across all channels. Like fMRI, the TRF RDMs were compared to model RDMs (see S3 Fig). First, model and data RDMs were vectorised (lower triangular) and then correlated (Spearman $r$) with each other for each time point and each subject. The statistical significance of the mean correlation above zero was tested with right-tailed $t$ test, FDR-corrected for 100 time points.

**TRF-fMRI fusion.** Representational similarity analysis was used to combine EEG and fMRI data [45,46]. The TRF RDMs for 100 time points (0 to 800 ms) were correlated (Spear-man $r$) with the 360 fMRI RDMs. Prior to correlations, the TRF RDMs were averaged across subjects to reduce noise in the data. Furthermore, partial correlation (Spearman $r$) was used, and the effect of task and background speech was controlled for when fusing dialogue TRFs and fMRI, and the effect of task and dialogue was controlled for when fusing background speech TRFs and fMRI. The statistical significance of the mean correlation above zero was tested with right-tailed $t$ tests, FDR-correction was applied for time points, ROIs, and models (task and dialogue/background speech).

## Supporting information

**S1 Video. Video illustration of the full-time series TRF-fMRI fusion and results for the dialogue stream.** TRFs were separately estimated for the dialogue streams for each combina-tion of semantic coherence and audiovisual quality and EEG channel. Average TRFs are dis-played for frontocentral electrodes in the middle column (attend speech: red, ignore speech: blue). We constructed TRF RDMs for each time point by correlating each EEG channel TRF pairwise across conditions and participants. Similar fMRI RDMs were constructed based on SVM decoding between the 16 condition pairs from fMRI data. Thus, we constructed similar RDMs for the EEG and the fMRI, allowing us to fuse information from both datasets by corre-lating vectorised TRF RDMs with fMRI RDMs, controlling for task and opposite speech stream TRF RDMs (see Fig 3C). To identify fMRI activations which corresponded to TRF RDMs at different time points, we conducted one-sample $t$ tests ($df$ = 18, FDR corrected) aver-aged across HCP parcellation ROIs. Code and processed EEG and fMRI data used to generate the video are archived on the Open Science Framework; HTTPS://DOI.ORG/10.17605/OSF. IO/AGXTH.
(AVI)

**S2 Video. Video illustration of the full-time series TRF-fMRI fusion and results for the background stream.** TRFs were separately estimated for the background streams for each com-bination of semantic coherence and audiovisual quality and EEG channel. Average TRFs are displayed for frontocentral electrodes in the middle column (attend speech: red, ignore speech: blue). We constructed TRF RDMs for each time point by correlating each EEG channel TRF pairwise across conditions and participants. Similar fMRI RDMs were constructed based on SVM decoding between the 16 condition pairs from fMRI data. Thus, we constructed similar RDMs for the EEG and the fMRI, allowing us to fuse information from both datasets by corre-lating vectorised TRF RDMs with fMRI RDMs, controlling for task and opposite speech stream TRF RDMs (see Fig 3C). To identify fMRI activations which corresponded to TRF RDMs at dif-ferent time points, we conducted one-sample $t$ tests ($df$ = 18, FDR corrected) averaged across HCP parcellation ROIs. Code and processed EEG and fMRI data used to generate the video are

archived on the Open Science Framework; HTTPS://DOI.ORG/10.17605/OSF.IO/AGXTH.
(AVI)

**S1 Data. Data frames to reproduce plots displayed in Fig 2B–2D.** We used the IBM 18 SPSS Statistics 25 (IBM SPSS, Armonk, New York, USA), UNIANOVA method, and EMMEANS to derive cell means and error terms. In Fig 2B, dependent variable was SER and factors were Segment and Condition. In Fig 2C, dependent variable was Percent_signal_change and factors were Line. In Fig 2D (left), dependent variable was SER and factors were Line and Condition. In Fig 2D (middle), dependent variable was Decoding_accuracy and factors were Line. In Fig 2D (right), dependent variable was Beta_weight_trimmed and factors were Line, note that the untrimmed Beta_weight is also provided.
(XLSX)

**S2 Data. Data frames to reproduce plot displayed in S1 Fig.** We used the BM 18 SPSS Statistics 25 (IBM SPSS, Armonk, New York, USA), UNIANOVA method, and EMMEANS to derive cell means and error terms. The dependent variable was SER and factors were Coherence, Auditory_quality, and Visual_quality.
(XLSX)

**S3 Data. Data frames to reproduce plot displayed in S2 Fig.** We used the IBM 18 SPSS Statistics 25 (IBM SPSS, Armonk, New York, USA), UNIANOVA method, and EMMEANS to derive cell means and error terms. In S2 Fig (left), the dependent variable was Mean_performance_percent and factors were Coherence, Auditory_quality, and Visual_quality. In S2 Fig (middle), the dependent variable was Mean_performance_percent and factors were Coherence, Auditory_quality, and Visual_quality. In S2 Fig (left), the dependent variable was Mean_performance_percent and factors were line.
(XLSX)

**S1 Text. Supplementary results text.**
(DOCX)

**S1 Fig. SER-accuracy changed depending on Attentional task, Semantic Coherence, Auditory, and Visual Quality.** SER accuracy was estimated using SER-correlation difference scores (see 4.9.4, for details; here the difference between the attend task and the ignore task is displayed). Error bars denote ± SEM. Abbreviations: p, poor; g, good; inco, incoherent; co, coherent; V, visual quality. Code and processed EEG data used to generate this figure are archived on the Open Science Framework; HTTPS://DOI.ORG/10.17605/OSF.IO/AGXTH. Data frames are available in S2 Data.
(TIFF)

**S2 Fig. Behavioural results.** Performance (percentage of correct answers) for the attend speech task (left), ignore speech task (middle). The rightmost plot shows the percentage of correct answers for the attend speech task by number of lines in the dialogue. Error bars denote ± SEM. Abbreviations: p, poor; g, good; inco, incoherent; co, coherent; V, visual Quality. Data frames are available in S3 Data.
(TIFF)

**S3 Fig. Representational dissimilarity (RDM) model correlations for the temporal response function (TRF) RDM time series separately for dialogue and background speech streams.** We constructed all possible main effect and interaction RDM models and correlated them with the speech stream RDM time series (see section 4.11). The attentional task (Att. vs. ign.) model yielded significant correlations (FDR corrected) for both the dialogue and the

background stream TRF-RDM time series. No other model (including those not displayed here) yielded significant FDR-corrected correlations. Here, we display some of the models that yielded reliable uncorrected, $p < 0.05$ correlation. Comparing the plots displayed with the TRF-fMRI fusion displayed in Fig 4 and S1–S2 Videos reveals that early effects for the dialogue stream (i.e., 0–150 ms) were affected by an interaction between Att. vs. ign. × Auditory Quality × Visual Quality. The effects between 250–450 were affected by a main effect of Visual Quality, and interactions between Att. vs. ign. × Auditory Quality, Att. vs. ign. × Coherence × Auditory Quality and Coherence × Auditory Quality × Visual Quality. The interactions that occurred around ca. 700 ms were affected by Att. Vs. ign. × Auditory Quality × Visual Quality. For the background speech, the early effects (around 0–50 ms) were affected by a Coherence × Visual Quality and the late (around 300 ms) by Coherence × Auditory Quality × Visual Quality. Code and processed EEG and fMRI data used to generate this figure are archived on the Open Science Framework; HTTPS://DOI.ORG/10.17605/OSF.IO/AGXTH.
(TIFF)

**S4 Fig. TRF-fMRI fusion results averaged for 10 different a priori selected regions of interest (ROIs).** We performed TRF-fMRI fusion to reveal when there were correspondences between the information structure of the TRFs and the fMRI data (for details see main text, Fig 4 and S1–S2 Videos). Here, we display the TRF-RDM-fMRI correlation time series averaged in 10 a priori selected ROIs of the HCP parcellation (for ROI names, see [60]). We selected A1 and PBelt based on the meta-analysis on selective attention effects for speech stimuli in the auditory cortex [87]. The STS ROIs (TPoJ1, STSdp, STSda) were chosen because these regions showed the strongest effects of selective attention in [38]. The IFG ROIs [44,45] were selected because they were described as central nodes of the speech control networks in our previous fMRI studies [38–40,42]. V1 and MT was chosen because the presented speech was audiovisually presented, i.e., contained moving visual speech. The dorsolateral prefrontal region p9-46v was chosen because this region was shown to change its connectivity with the auditory cortex during selective attention to speech in our previous analyses on the fMRI data [38]. Correlations are displayed for the dialogue and the background sound stream as well as the attentional task model (temporally demeaned). Code and processed EEG and fMRI data used to generate this figure are archived on the Open Science Framework; HTTPS://DOI.ORG/10.17605/OSF.IO/AGXTH.
(TIFF)

## Acknowledgments

We would like to thank Viivi Kanerva, Elisa Sahari, and Artturi Ylinen for help with the gathering of the EEG and fMRI data and Ilkka Muukkonen for consultation with the decoding analyses.

## Author Contributions

**Conceptualization:** Patrik Wikman, Matti Laine, Kimmo Alho.

**Data curation:** Patrik Wikman, Eetu Sjöblom.

**Formal analysis:** Patrik Wikman, Viljami Salmela, Eetu Sjöblom.

**Funding acquisition:** Patrik Wikman, Matti Laine, Kimmo Alho.

**Investigation:** Patrik Wikman, Viljami Salmela.

**Methodology:** Patrik Wikman, Viljami Salmela, Eetu Sjöblom, Miika Leminen.

**Project administration:** Kimmo Alho.

**Resources:** Miika Leminen.

**Software:** Patrik Wikman, Viljami Salmela, Eetu Sjöblom.

**Supervision:** Patrik Wikman, Kimmo Alho.

**Validation:** Patrik Wikman.

**Visualization:** Patrik Wikman, Viljami Salmela.

**Writing – original draft:** Patrik Wikman, Viljami Salmela, Eetu Sjöblom, Miika Leminen, Matti Laine, Kimmo Alho.

**Writing – review & editing:** Patrik Wikman, Viljami Salmela, Eetu Sjöblom, Miika Leminen, Matti Laine, Kimmo Alho.

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
