## [Editor Report · Decision Letter 0]

21 Sep 2023

Dear Dr Wikman, 

Thank you for submitting your manuscript entitled "Selective attention to audiovisual speech routes activity through recurrent feedback-feedforward loops between different nodes of the speech network" for consideration as a Research Article by PLOS Biology.

Your manuscript has now been evaluated by the PLOS Biology editorial staff as well as by an academic editor with relevant expertise and I am writing to let you know that we would like to send your submission out for external peer review.

Once your full submission is complete, your paper will undergo a series of checks in preparation for peer review. After your manuscript has passed the checks it will be sent out for review. To provide the metadata for your submission, please Login to Editorial Manager (https://www.editorialmanager.com/pbiology) within two working days, i.e. by Sep 23 2023 11:59PM.

Kind regards,

Christian

Christian Schnell, PhD

Senior Editor

PLOS Biology

cschnell@plos.org

---

## [Decision Letter · Decision Letter 1]

8 Dec 2023

Dear Dr Wikman,

Thank you for your patience while your manuscript "Selective attention to audiovisual speech routes activity through recurrent feedback-feedforward loops between different nodes of the speech network" went through peer-review at PLOS Biology. Your manuscript has now been evaluated by the PLOS Biology editors, an Academic Editor with relevant expertise, and by several independent reviewers.

In light of the reviews, which you will find at the end of this email, we are pleased to offer you the opportunity to address the concerns from the reviewers in a revision that we anticipate should not take you very long. We will then assess your revised manuscript and your response to the reviewers' comments with our Academic Editor aiming to avoid further rounds of peer-review, although might need to consult with the reviewers, depending on the nature of the revisions.

We expect to receive your revised manuscript within 1 month. Please email us (plosbiology@plos.org) if you have any questions or concerns, or would like to request an extension. We are happy to extend the deadline given the upcoming holidays and the fact that you may need to include some additional analyses.

**IMPORTANT - SUBMITTING YOUR REVISION**

*Resubmission Checklist*

*Published Peer Review*

*PLOS Data Policy*

Please note that as a condition of publication PLOS' data policy (http://journals.plos.org/plosbiology/s/data-availability) requires that you make available all data used to draw the conclusions arrived at in your manuscript. If you have not already done so, you must include any data used in your manuscript either in appropriate repositories, within the body of the manuscript, or as supporting information (N.B. this includes any numerical values that were used to generate graphs, histograms etc.). For an example see here: http://www.plosbiology.org/article/info:doi%2F10.1371%2Fjournal.pbio.1001908#s5

*Blot and Gel Data Policy*

Sincerely,

Christian

Christian Schnell, PhD

Senior Editor

PLOS Biology

cschnell@plos.org

REVIEWS:

Reviewer's Responses to Questions

Do you want your identity to be public for this peer review?

Reviewer #1: Yes: Brigitta Toth

Reviewer #2: Yes: E Schröger

Reviewer #1: Review 

Title: Selective attention to audiovisual speech routes activity through recurrent feedback- feedforward loops between different nodes of the speech network

PBIOLOGY-D-23-02396

The present study combined electrophysiology and functional brain imaging to explore the cortical regions involved in top-down modulations during multi-talker situations. The task design very ambitiously aimed to simultaneously manipulate the focus of attention, semantic coherence (context), and attentional demand (auditory and visual quality). They demonstrated neural tracking of attended speech supported by the ventral auditory stream. Furthermore, the attentional effect on the accuracy of speech envelope reconstruction depended on the course of the dialogue, suggesting that predictive and attentional mechanisms interact via neural tracking. 

Overall, I support the publication of this study, with a few suggestions. 

1) The paragraph explaining the experimental design in lines 89-99 needs to mention the hypothesis or describe the rationale behind using each type of experimental manipulation. It is necessary to include a reason and related hypothesis for using semantically coherent or incoherent dialogues. The introduction could benefit from explaining this. 

2) Please briefly define the term representational dissimilarity matrix at the first mention in the text.

3) Factorial design means that all subjects performed the two tasks (selective attention) under all types of semantic coherence and attention demand conditions. Is that correct? Please disambiguate in the text. 

4) Was there enough statistical power to test all the effects of attention demand, selective attention, and context? 

5) Typically, neural tracking is used to compare the effect of attention on the attended versus background stream. Still, the present design allows the authors to compare the task effect of selective attention on ignored versus attended speech. Was the same sensory input compared in two task conditions, with the same dialogue presented in different conditions? Is there a confound of repetition?

6) Was there a significant increase in neural tracking of the speech from attended dialogue compared to the background stream?

7) Lines from 164: "First, we examined within-line effects, that is, whether SER accuracy changed within the line (lines were divided into four equal length segments)"; "Line of the speech stream (1-7), and Segment of the line (1-4)". Regardless of which line it was, the segment order affected the accuracy of the SER. Is that correct? Why was segment order analyzed instead of the effect of the line? Why do the authors assume that SER accuracy changes during the dialogue are more representative through the segment than the lines?

8) What does the timescale of the TRF response reflect? It is mentioned that it can be conceptualized as an ERP to a continuous variable, but what continuous variable the authors refer to needs to be clarified. Is it the speech signal?

9) In line 264, it is stated that selective attention enhanced the TRF for the dialogue stream, but it would be more precise to say what was compared here (the dialogue vs. background stream)

10) It is hard to interpret the difference in conceptualization of the results for the attentional differences in TRF-RDM analysis and the one that compares TRF between ignored and attended speech. Please elaborate on it, at least in the discussion section.

11) In Fig. 4C, the fMRI contrast shows the most substantial contrast on the visual cortex. How did it come about? That is the source of the most prominent computational difference between the two tasks (visual vs. auditory). It suggests that this analysis only reflects the most cortical differences between the two tasks, but in line 372, the authors state that the analysis is not due to the main effect of the task. Thus, it needs to be discussed. 

12) Correlation between TRF and fMRI RDM arose in 16 ms and 30-78 ms for the dialogue stream and about 8 ms for the background stream. This is contradicting, given that fMRI has a really low temporal resolution. What these timescales refer to needs to be defined more clearly (EEG TRF response?). Also, the fact that TRF and fMRI datasets were collected from unidentical subjects needs to be made explicit in the main text, not just in the methods section.

13) I suggest adding a reference that also shows the selective attention effect of unattended speech in a broad cortical network at lines 420, 426 and 436: Tóth, B., Farkas, D., Urbán, G., Szalárdy, O., Orosz, G., Hunyadi, L., ... & Winkler, I. (2019). Attention and speech-processing-related functional brain networks are activated in a multi-speaker environment. PLoS One, 14(2), e0212754. (mi lendület cikkünk PLOS one )

Minor comments

1) Fig. 2C shows the temporal changes in attentional modulation. It needs to be clarified what the color coding on the cortical surface represents. The label for the y-axis needs to be included in the diagram below, and the definition of primary speech network should be stated at least in the captions. 

2) In line 191, the first time the slow temporal effect is mentioned, it would be helpful to define it (line effect - slow temporal effect). 

3) In Fig. 3A, the text above the TRF for each EEG channel (bottom panel) is too small to read. 

Reviewer #2: Wikman and colleagues report EEG and fMRI data from human participants selectively attending to audiovisual speech of two talkers (dialogue) with concurrent background speech (cocktail party situation). Participants either attended to the (foreground) speech (solving questions after every dialog) or attended to a visual cross (i.e. ignoring the speech and answering questions related to the turning of the cross). The quality of the sound and visual stimuli (speech and video of talkers) was manipulated (good vs. poor). The EEG results showed that attention increased the neural tracking of speech. Moreover, the accuracy of the speech envelope reconstruction was modulated over time of the dialogue. This was mirrored by the fMRI results. Applying temporal response function analysis to the EEG data (yielding temporal information similar to ERP analysis) revealed an effect of attending to the dialog in a window around 75 ms and another around 300 ms. Multivariate representational similarity analysis on the temporal response functions (yielding a pointwise temporal response function representational dissimilarity matrix along the time course of each stream) revealed an effect of the effect of attentional task (attend versus ignore the sounds). There was an effect for the dialogue but also for the background speech. The fMRI data revealed for this contrast effects in many (also non-auditory) regions of the brain. Finally, performing EEG-fMRI fusion revealed a series of back (to auditory) and forth (from auditory) correlations in a temporal window of few hundred of ms for the dialogue stream (first, at less than 20 ms after changes in the sound envelope in dorsolateral and dorsomedial frontal regions, which were succeeded by correlations in auditory regions, which were followed by correlations in anterior temporal areas, and then followed by correlations back to auditory and non-auditory regions). Also for the background stream back-forth correlations were observed, however, starting at a latency of around 450 ms. While the EEG and fMRI analysis replicated (and partly extended) existing knowledge of attention effects in cocktail party situation, the EEG-fMRI fusion data hint at feedforward-feedback informational flow between auditory areas and areas associated with speech and executive functions.

In my opinion, this is a highly innovative study using a clever experimental design and a sophisticated analysis of the EEG and fMRI data. The majority of within auditory attention research and between audition versus vision attention research results were confined to either showing attention effects in time or in space. A chronometry of the brain's network of within and between modality auditory attention effects has (to my knowledge) never be performed. Thus the present results are a huge achievement in the classic but still very timely topic of auditory attention. The results are convincingly framed within current ideas about neural circuit learning / selection and recurrent low-level-sensory-high-level-non-sensory loop processing (e.g. by Michael Kilgard and Erkki Schröger).

Though the manuscript Is well written and is in a very good shape, I have some comments:

1)

Discussion

As Wikman a colleagues somehow open a new avenue in auditory attention research, they might shortly set their findings into a somewhat broader context, that is, to different but still related research fields. I have three possible "candidates" in mind, but there might be more appropriate ones. In psychology Jones and Riess were forerunners in studying an conceptualizing the dynamics of attending to sound (though their dynamics relates to larger temporal scale and to the entrainment effects set by the context; this could be related to the dialogue lines findings), e.g. Large, EW; Jones, MR (1999): The dynamics of attending: How people track time-varying events. In: Psychological Review 106 (1), S. 119-159. DOI: 10.1037/0033-295X.106.1.119. In neurobiology, for example, Jonathan Fritz, Mounya Elhilali, and Shihab Shamma were pioneers in showing auditory attention effects on the respective receptive field can be rapidly adapt; this could be taken as a possible biological implementation of some of the results; e.g., Fritz, JB; Elhilali, M, Shamma, SA. (2007): Adaptive changes in cortical receptive fields induced by attention to complex sounds. In: Journal of neurophysiology 98 (4), S. 2337-2346. DOI: 10.1152/jn.00552.2007. In computational modelling the auditory attention effect in cocktail party situations the EEG temporal response function approach has been modelled with recurrent neural network and reinforcement learning methods.; e.g. Geravanchizadeh, Masoud; Roushan, Hossein (2021): Dynamic selective auditory attention detection using RNN and reinforcement learning. In: Scientific reports 11 (1), S. 15497. DOI: 10.1038/s41598-021-94876-0.

2)

P5 lines 134-137

This paragraph consists of one sentence (which is not common) and it is difficult to understand. Please, split in two sentences.

3)

P10 Fig. 3

The font size is too small in several parts of the Fig (e.g."att", "ign", numbers)

This partly applies to Fig. 4 as well.

4)

P12, l 355

The link to www.mv.helsinki.fi/home/jkaurama/vdialog/ did not work for me (the other video links worked)

5)

P13, l 364

"there was initially correlations" should (probably) read as "there were initially correlations".

6)

P26, line 793

"the data was bandpass filtered (0.5-10 Hz)" 

Please, add filter characteristics, if available.

7)

P29, line 867

"approximate effect sizes were applicable." should probably read as "approximate effect sizes where applicable."

---

## [Editor Report · Decision Letter 2]

10 Jan 2024

Dear Dr Wikman,

Thank you for your patience while we considered your revised manuscript "Selective attention to audiovisual speech routes activity through recurrent feedback-feedforward loops between different nodes of the speech network" for publication as a Research Article at PLOS Biology. This revised version of your manuscript has been evaluated by the PLOS Biology editors and the Academic Editor.

Based on our Academic Editor's assessment of your revision, we are likely to accept this manuscript for publication, provided you satisfactorily address the following data and other policy-related requests:

* We would like to suggest a different title to improve readability: 

Attention to audiovisual speech shapes neural processing through feedback-feedforward loops between different nodes of the speech network

* Please add the links to the funding agencies in the Financial Disclosure statement in the manuscript details

* Please clarify the last sentence in your ethical approval statement which currently reads "Written informed consent was obtained from the sharing of anonymized data. for the publication of any potentially identifiable images or data included in this manuscript."

* Please provide a blurb which (if accepted) will be included in our weekly and monthly Electronic Table of Contents, sent out to readers of PLOS Biology, and may be used to promote your article in social media. The blurb should be about 30-40 words long and is subject to editorial changes. It should, without exaggeration, entice people to read your manuscript. It should not be redundant with the title and should not contain acronyms or abbreviations.

DATA POLICY:

Regardless of the method selected, please ensure that you provide the individual numerical values that underlie the summary data displayed in the following figure panels as they are essential for readers to assess your analysis and to reproduce it: 2B, 2C, 2D, S1, and S2. 

* Please provide a DOI for your OSF repository, so that the repository is citable and versioned for your paper (https://help.osf.io/article/220-create-dois).

We expect to receive your revised manuscript within two weeks. 

*Published Peer Review History*

*Press*

Sincerely,

Christian

Christian Schnell, PhD

Senior Editor,

cschnell@plos.org,

PLOS Biology

---

## [Editor Report · Decision Letter 3]

30 Jan 2024

Dear Dr Wikman,

Thank you for the submission of your revised Research Article "Attention to audiovisual speech shapes neural processing through feedback-feedforward loops between different nodes of the speech network" for publication in PLOS Biology. On behalf of my colleagues and the Academic Editor, Manuel Malmierca, I am pleased to say that we can in principle accept your manuscript for publication, provided you address any remaining formatting and reporting issues. These will be detailed in an email you should receive within 2-3 business days from our colleagues in the journal operations team; no action is required from you until then. Please note that we will not be able to formally accept your manuscript and schedule it for publication until you have completed any requested changes.

PRESS

Sincerely, 

Christian

Christian Schnell, PhD, PhD

Senior Editor

PLOS Biology

cschnell@plos.org